# ALTAI Tool for Assessing AI-Based Technologies: Lessons Learned and Recommendations from SHAPES Pilots

**DOI:** 10.3390/healthcare11101454

**Published:** 2023-05-17

**Authors:** Jyri Rajamäki, Fotios Gioulekas, Pedro Alfonso Lebre Rocha, Xavier del Toro Garcia, Paulinus Ofem, Jaakko Tyni

**Affiliations:** 1Unit W, Laurea University of Applied Sciences, 02650 Espoo, Finland; 25th Regional Health Authority of Thessaly & Sterea, 41110 Larissa, Greece; 3CINTESIS@RISE, Department of Behavioural Sciences, School of Medicine and Biomedical Sciences (ICBAS), University of Porto, 4200-450 Porto, Portugal; 4Computer Architecture and Networks Group, School of Computer Science, University of Castilla-La Mancha, Paseo de la Universidad, 4, 13071 Ciudad Real, Spain

**Keywords:** healthcare, aging, artificial intelligence, assessment, ethics, ALTAI, SHAPES Project

## Abstract

Across European countries, the SHAPES Project is piloting AI-based technologies that could improve healthcare delivery for older people over 60 years old. This article aims to present a study developed inside the SHAPES Project to find a theoretical framework focused on AI-assisted technology in healthcare for older people living in the home, to assess the SHAPES AI-based technologies using the ALTAI tool, and to derive ethical recommendations regarding AI-based technologies for ageing and healthcare. The study has highlighted concerns and reservations about AI-based technologies, namely dealing with living at home, mobility, accessibility, data exchange procedures in cross-board cases, interoperability, and security. A list of recommendations is built not only for the healthcare sector, but also for other pilot studies.

## 1. Introduction

According to forecasts, by 2050, Europe will have approximately 130 million inhabitants, of which almost a third will be 65 years or older [1]. Studies have shown the benefits of ageing in place, both in the home and community. Nevertheless, older people living at home and in the community also need healthcare services [1,2,3,4,5].

Ambient Assisted Living (AAL) technologies—which range from cheap devices to complex Artificial Intelligence (AI) and Machine Learning (ML) technologies—aim to take care of the growing number of older people [6]. They are purposed to detect things and events that would otherwise go unnoticed. AI and ML can enable and support disease prevention in the healthcare sector in parallel to disease treatment. According to recent studies, wearable devices with AI can save up to 313,000 people across the EU, around EUR 50 million, and about 300 million hours of healthcare professionals’ working time [7].

### 1.1. Artificial Intelligence in Healthcare

The quality of technologies, especially AI-based technologies, is based on learning systems that use personal data and algorithms, often non-public trade secrets [8]. Additionally, the public guidelines have no coherent and consolidated approaches for technology adoption by AI developers [9] and healthcare professionals. Recent studies have highlighted that healthcare professionals’ (physicians and nurses) perceptions are a critical issue in adopting AI technologies (such as AAL) in healthcare delivery; the negative perception of the trustworthiness of AI is tantamount to a rejection of AI [10,11,12,13].

Shinners et al. developed and validated a survey tool (questionnaire) to investigate healthcare professionals’ perceptions towards AI. The questionnaire comprised 11 items bordering on respondent demographics such as age and other items centered on perceptions of AI’s impact on the future role of healthcare professionals and their professional preparedness. Shinners et al. conducted a reliability test of the survey tool. They found that the use of AI had a significant effect on the perception of healthcare professionals regarding their preparedness and future role in the healthcare sector. This initial outcome showed that the survey instrument helps explore AI perceptions [14].

Another study with different health fields (ophthalmology, dermatology, radiology, oncology) demonstrated that the clinicians’ acceptability of AI technologies was determined by how much the AI’s performance surpasses the average performance threshold of human specialists. In ranking the perceived advantages of AI, participants believed improved diagnostic confidence that borders on trustworthiness is only subsequent to improved patient access to disease screening. Reduced time spent on monotonous tasks appeared third in the ranking. Despite these perceived advantages of AI, concerns remain, as the study further indicated. The top-ranked among these concerns raised by respondents are the divestiture of healthcare to technology and data companies, medical liability owing to machine error, and a decrease in reliance on medical specialist diagnosis and treatment counselling. The second concern highlights the need for trustworthy AI [15].

In a French study, Laï et al. found that AI is considered a myth requiring debunking among healthcare professionals. These professionals find AI tools useful in delivering healthcare. However, they see less and less incorporation of AI in their daily practice. This outcome might not be unconnected with healthcare industry partners’ perceived legal bottlenecks surrounding access to individual health data, thus hampering AI adoption. For this study, AI acceptability is often influenced by accountability, hence the concern about the possible harm that AI clinical tools might cause patients [16].

In addressing accountability, Habli et al. highlighted the role of moral accountability of AI harm for patients and safety assurance to protect patients against resulting harm caused by AI tools. The authors called for a review of the current practice of accountability and safety in using AI tools for decision making. To mitigate AI potential harm to patients and for an encompassing evaluation of AI tools, Habli et al. argued that stakeholders such as AI developers and systems safety engineers need to be considered when assessing moral accountability for patient harm [17].

Transparency is also a factor against AI tool adoption. Markus et al. proposed a framework that guides AI developers and practitioners in choosing between different categories of explainable AI methods that may be adopted. Adopting explainability in creating trustworthy AI creates a demand for explainability, determining what should be explained about the AI tool. The authors argued that applying explainable modelling could help make AI more trustworthy, even as the perceived benefits require proof in practice, especially in healthcare [18].

Other studies pointed out the patients’ perspectives. For example, Nichols et al. showed that patients who participated in a survey investigating confidence levels between AI and clinician-assisted interpretation of radiographic images favored the clinician-led radiograph interpretation. In addition, based on participant demographics, younger and more educated patients tend to favor AI-assisted image interpretation [19]. Another piece of research revealed that security and privacy are key factors that are perceived to influence medical assistive technologies, including AI technologies. The authors found that young and middle-aged individuals are more concerned about security and privacy standards than the ailing elderly population who participated in the study [20].

### 1.2. Ethical Framework for Trustworthy AI

Since the integration of AI-assisted technologies in healthcare interacts with the perceptions, acceptability, accountability, transparency, explainability, privacy, security, and literacy of both patients and physicians, the development of new ethical frameworks is required [21]. Morley et al. [22] proposed three ethical levels to discuss AI in healthcare: (a) epistemic, related to misleading, incomplete, or unexamined evidence; (b) normative, related to unjust outcomes and transformative effects; and (c) dealing with traceability [22].

Nevertheless, there is still no consensus because what can be effective and desirable for society may not necessarily be desirable for an individual [23]. At the same time, some stakeholders agree on ethical issues of AI at the principal level. In contrast, others disagree on how the principles are interpreted, why they are important, which issues, domains, or actors they apply to, and how they should be implemented [24].

It is also unclear how the principles should be prioritized, how conflicts between ethical principles should be resolved, who should monitor AI’s ethics, and how different parties can comply with the principles. Different interpretations of the principles only become apparent when the principles or concepts are tested in their context, which is important to understand [25]. According to Yin et al., these results indicate a gap between creating principles and practical implementation [26].

In the preceding context, effective collaborations with all stakeholders (IT developers, healthcare professionals and patients, universities and research centers, and governments) in healthcare are mandatory to build critical structures that promote trust in AI technologies. These could include appropriate safeguards to protect and maintain patient agency, clinical decision making and support for medical diagnosis, checklists and guidelines for AI developers, and user and technical requirements for professionals and services [27].

The legitimacy of an AI implies that it is lawful in that it respects all applicable laws and regulations. The ethics of an AI means that the AI application complies with all known ethical principles, societal values, and technical requirements. There are several parallel global efforts towards achieving international standards (e.g., ISO/IEC 42001 AI Management System) [28], regulations (e.g., AI Act) [29], and individual–organizational policies relevant to AI and its trustworthiness features [30].

The European Commission’s High-Level Expert Group on Artificial Intelligence (AI HLEG) published the European Commission’s Ethical Guidelines for Trustworthy AI in 2019. These guidelines set standards for three main requirements when developing an AI system: legitimacy, ethics, and reliability [31]. These EU recommendations are based on human rights, which, according to the EU Treaties and the EU Charter, comprise values such as human dignity, equality, freedom, solidarity, justice, and civil rights. Developed by AI HLEG, the Assessment List for Trustworthy AI (ALTAI) is an assessment list that aims to safeguard against AI harms in healthcare and requires a shared global effort. The AI HLEG report offers room for localized solutions that may not foster trustworthy AI globally [32,33]. Furthermore, there is an online version of this tool [34].

## 2. Materials and Methods

### 2.1. Context, Purposes, and Methodology

The current study is integrated into the SHAPES Project work group “SHAPES Legal, Ethics, Privacy and Fundamental Rights Protection,” led by Laurea University of Applied Sciences (Laurea) [10]. The objectives are (1) to find a theoretical framework focused on AI-assisted technology in healthcare for older people living at home; (2) to assess the SHAPES pan-European pilots by the ALTAI tool; and (3) to derive ethical recommendations regarding AI-based technologies for ageing and healthcare. The study adopts multiple case study methodologies [32,35] with the scope to explore the AI-based technology in healthcare for older people in a fine-grained, in-depth, and contextualized way; it also compares cases, maps their differences and similarities, and identifies emerging patterns.

### 2.2. Recruitment, Collection, and Analysis and Validation

The study was initiated by Laurea’s researchers through emails to all SHAPES partners (consortium) targeting to explain the study’s framework (context, purposes, methodology) and invite SHAPES Pilots’ leaders. In this email, the researchers have defined five criteria to enable the participation in the study: (1) members of SHAPES consortium; (2) partners directly engaged in SHAPES Pilots as researchers, IT developers, or healthcare providers; (3) partners who participate in all the study’s phases; (4) pilots that are testing AI-based technology to provide healthcare in the home; and (5) pilots approved by ethical committees. Three of the seven SHAPES Pilots met all these criteria.

Laurea’s researchers, supported by the leaders of the SHAPES Pilots selected, have produced a literature review to define a theoretical framework for the study. They limited the research to two databases (PubMed and Scopus). They used different articulations between five keywords (Ageing, Healthcare, Artificial Intelligence, Ethics, and ALTAI). The results are presented in the Introduction section.

The leaders (coordinators) of the SHAPES Pilots selected have completed the development of the online version of the ALTAI tool. The ALTAI tool is a list of requirements (seven) that evaluate AI trustworthiness: (1) Human Agency and Oversight; (2) Privacy and Data Governance; (3) Technical Robustness and Safety; (4) Accountability; (5) Transparency; (6) Societal and Environmental Well-being; and (7) Diversity, Non-discrimination, and Fairness. At the end of each section, a self-assessment is performed on a 5-level assessment scale (Likert scale): ‘non-existing’ corresponds to 0; ‘completely inadequate’ to 1; ‘almost adequate’ to 2; ‘adequate’ to 3; and ‘fully adequate’ corresponds to 4 [34].

The ALTAI is an interactive self-assessment tool; different questions are suggested according to the answers provided. All the questions can be consulted in the tool guideline: “ETHICS GUIDELINES FOR TRUSTWORTHY AI: High-Level Expert Group on Artificial Intelligence” [31] (pp. 26–31). Each question has a glossary and examples from the Ethics Guidelines for Trustworthy AI. The questions are color-coded to describe the features of the AI system; blue questions will contribute to generating the recommendations, and red questions aim to self-assess compliance with the requirement/blue answer. When the ALTAI questionnaire is completed, a visual graphic (diagram) with the evaluation and a recommendations list is generated automatically (Figure 1). The following link accesses the platform: https://altai.insight-centre.org/Identity/Account/Login (accessed on 15 March 2023).

Laurea’s researchers have conducted three online workshops, one per SHAPES Pilot selected. The workshops were targeted at the partners that directly engaged in the pilots (researchers, IT developers/technology owners, and healthcare providers). These workshops aimed to present the ALTAI self-assessment and literature review results, collect data regarding the pilots’ design and lessons learned, and inquire about the partners’ perspectives on ALTAI assessment, namely the recommendations.

A triangulation analysis was performed to understand similarities and differences between the SHAPES Pilots (case studies) and to validate the lessons learned and the list of final recommendations defined related to the ALTAI recommendations. Figure 2 depicts the aforementioned steps of the study.

### 2.3. Limits and Ethics

This study involved a small number of participants, which reduces the recommendations’ scalability; one person (leader/coordinator) per pilot who represents the team engaged in the pilots’ engagement. There was no requirement for ethical approval for this study because all participants are members of a consortium, the SHAPES Project (Grant Agreement No. 857159), which already has an ethics and GDPR framework required by the funding program (The European Union’s Horizon 2020 research and innovation program). Each SHAPES Pilot is supported by the corresponding ethical approval from national ethical committees.

## 3. Results

### 3.1. SHAPES Pilots

The SHAPES Project is building and piloting a large-scale, EU-standardized open platform that integrates digital solutions and sociotechnical models to promote long-term healthy and active ageing and maintain a high-quality standard of life. Using digital solutions (and respective sociotechnical models) in the community and home (APPs, Voice Assistants, Sensors, Smart Wearables, Software, Medical Devices), older people, caregivers (formal and informal), and healthcare providers could be better integrated. Healthcare delivery could have a higher impact (e.g., preventive care, self-care, reducing costs and hospitalizations) [10].

This ‘vision’ is being tested by the SHAPES Pan-European Pilot Campaign, which includes building and demonstrating in a real-life context several digital solutions and sociotechnical models across different European countries and users, namely older people, informal caregivers, formal caregivers (e.g., nursing homes), and healthcare providers (e.g., health authorities). The pilot campaign is divided into seven thematic or “pilot themes” (PTs) to cover many domains related to healthy ageing, independent living, and integrated care: (PT1) Smart Living Environment for healthy ageing at Home; (PT2) Improving In-Home and Community-based Care; (PT3) Medicine Control and Optimization; (PT4) Psycho-social and Cognitive Stimulation Promoting Wellbeing; (PT5) Caring for Older Individuals with Neurodegenerative Diseases; (PT6) Physical Rehabilitation at Home; and (PT7) Cross-border Health Data Exchange [10].

All the SHAPES Pilots were expected to test AI-based technologies; however, only three have fitted in with all recruitment criteria at the time of this study. There were two key reasons for not participating: they had not yet obtained ethics approval and did not complete all phases of the study. Within this context, this study considered only three pilot themes (i.e., PT5, PT6, and PT7). In the Table below, these pilots are described in four topics: (1) Persona and Use Case, (2) Digital Solution, (3) the AI ‘role’, and (4) SHAPES Protocol (Table 1).

### 3.2. ALTAI Assessment

Based on the answers provided by the pilots’ partners, a cross-case analysis (Figure 3) was conducted to understand the similarities and differences between different pilots (case studies). Globally, the best result is obtained from the evaluation point of ‘transparency’ and the worst from ‘technical robustness and safety’. Moreover, each partner has received a list of recommendations (Appendix A).

Regarding human agency and oversight, three recommendations were given for more than one pilot. These recommendations were given to Pilots 6 and 7. They aim to promote the responsible use of AI systems by avoiding over-reliance on the system, preventing inadvertent effects on human autonomy, and providing appropriate training and oversight to individuals responsible for monitoring the system’s decisions. 

There were two pilots that got no recommendation for the technical robustness and safety requirement. Pilot 5 received five recommendations, which aim to identify and manage the risks associated with using AI systems, including potential attacks and threats, possible consequences of system failure or malfunction, and the need for ongoing monitoring and evaluation of the system’s technical robustness and safety.

Concerning the Privacy and Data Governance requirement, Pilot 5 received no recommendations for this. Pilots 6 and 7 were advised to ‘establish mechanisms that allow flagging issues related to privacy or data protection concerning the AI system’. Pilot 7 received four individual recommendations, which aim to ensure that privacy and data protection are considered throughout the lifecycle of the AI system, from data collection to processing and use, and that appropriate mechanisms are in place to protect individuals’ privacy rights.

Regarding the Transparency requirement, Pilots 5 and 6 were recommended to consider regularly surveying the users to inquire about their comprehension of the AI system’s decision-making process. Pilots 6 and 7 were advised in the case of interactive AI systems to consider informing the users that they are engaging with a machine. Pilot 6 was advised to take steps to continuously assess the quality of input data used by their AI systems, explain the decisions made or suggested by the system to end-users, and regularly survey users to ensure they understand these decisions.

The conformance to the Diversity, non-discrimination, and fairness requirements received most of the recommendations, a total of 43 recommendations, of which 17 recommendations were given to at least two pilots. These recommendations, which were given at least to two pilots, can be further divided into the following subcategories:Data and algorithm design: This includes recommendations related to the input data and algorithm design used in the AI system, such as avoiding bias and ensuring diversity and representativeness in the data, using state-of-the-art technical tools to understand the data and model, and testing and monitoring for potential biases throughout the AI system’s lifecycle.Awareness and education: It includes recommendations associated with the education of AI designers and developers about the potential for bias, the discrimination in their work, and the established mechanisms for flagging bias issues. They ensure that information about the AI system is accessible to all users, including those with assistive technologies.Fairness definition: This includes recommendations related to defining fairness and consulting with impacted communities to ensure that the definition is appropriate and inclusive. It also includes suggestions for establishing quantitative metrics to measure and test the meaning of fairness.Risk assessment: This includes recommendations relevant to assessing the possible unfairness of the AI system’s outcomes on end-users or subject’s communities and identifying groups that might be disproportionately affected by the system’s outcomes.

The Societal and Environmental Well-being requirement received 11 recommendations, of which one recommendation was given to all three pilots, and three recommendations were given to two pilots. All three pilots were advised to establish strategies to decrease the environmental impact of their AI system throughout its lifecycle and participate in contests focused on creating AI solutions that address this issue.

Regarding Accountability, all three pilots got a recommendation that suggest that if AI systems are used for decision-making, it is important to ensure that the impact of these decisions on people′s lives is fair, in line with uncompromisable values, and accountable. Therefore, any conflicts or trade-offs between values should be documented and explained thoroughly.

## 4. Discussion

### 4.1. Lessons Learned

SHAPES partners have acknowledged that the ALTAI tool is a reliable and easy-to-use self-assessment tool for Trustworthy AI-based technology in pilot studies such as SHAPES pilots, which are of interest in the market. Similar analyses [11,12] have reported the importance of the ALTAI tool to provide reliable recommendations towards the improvement of trustworthiness in AI solutions. Specifically, although authors in [11] have slightly altered the ALTAI tool questions in their study, their results’ analysis showed an awareness of some of the broader key areas of trustworthy factors as indicated by the ALTAI tool, such as accuracy of the performance of the AI solution. Moreover, the study revealed that the participating researchers were unsure whether their work was covered by the various definitions and how applicable the recommendations could be to their solutions. They recommended that a repository of ALTAI tool experiences along with examples of good practices would be the key to the tradeoff between AI complexity and the compliance with ALTAI’s trustworthiness recommendations. Similarly, the study in [12] reported that industry and public bodies should both be engaged to efficiently regulate the incorporation of the ALTAI tool or its future improvements in the development of AI solutions.

Despite ALTAI not being directly addressed to the market, the tool would allow corporations to compare results with others at a similar level of sophistication or in similar application domains or locations. Before an AI tool is released to the market and during its evaluation process, developers and researchers could adopt this ALTAI to increase the trustworthiness level of the AI tools. Effective use of the ALTAI tool and other assessment and rating schemes is crucial for achieving trustworthy governance in AI, promoting consumer confidence in AI, and facilitating its adoption.

Nevertheless, and as mentioned before, we also identified that some cases cannot be considered, such as a lack of regulations worldwide or in the EU that could also be used. This presents difficulty in convincing the AI tool providers to fully address the ALTAI findings. Furthermore, an independent assessor could offer valuable insights for improvement in spite of the fact that the ALTAI tool has the benefits of simplicity and ease of implementation. For future EU regulation around AI to be successful, there needs to be an ecosystem of organizations, auditors, and rating schemes developed to address the challenges created by the industry.

The SHAPES partners also have pointed out an ALTAI weakness regarding the relative risk of AI systems in the assessment process. While there is an ongoing debate about the best approach to risk assessment, it is essential to embed risk weighting fully in any evaluation of the appropriateness of the level of governance for an AI system.

It should be noted here that the spider diagram is automatically generated by the ALTAI tool as described before. The final scores produced by the spider diagram are associated with the number of recommendations. Precisely, a high score corresponds to a high number of recommendations. However, the ALTAI method developers have not published how each answer to the ALTAI questions contributes to the final scores shown in the spider diagram.

Furthermore, it is important to involve specific co-creation processes in the iterative design methodologies employed by most of the developers of AI solutions. Within these processes, the participation of different stakeholders such as end-users or associated institutions is considered (i.e., organizations or chambers that will use the AI technology, such as doctors, retail branches, etc.). Therefore, AI trustworthiness checks could be part of the co-creation activities. After the completion of each one of the associated co-creation steps, the ALTAI recommendations should be taken into account without of course degrading the fidelity of the performance of the AI technology under development. Obviously, some of the recommendations given by the ALTAI tool should be involved from the beginning of the development procedure, as a design principle. Moreover, some of the stakeholders, involved in the self-assessments, noted that it would be useful to know the level of the compliance with a trustworthy AI and ALTAI tool of publicly available machine learning models and datasets before their incorporation into their AI solutions and in order to not alter the performance of their designs. Additionally, it should be mentioned that the recommendations provided by the ALTAI tool regarding the opportunities for improvement are invaluable for organizations seeking to build a road map for the maturity of their governance [12].

### 4.2. Recommendations

Based on the Horizon Europe ethics self-assessment orientations [13], the ALTAI tool could become the standard for any Horizon Europe projects involving AI-based technology. However, the ALTAI recommendations seem extensive, repeated, and difficult to understand and use. The inability of the stakeholders to interpret the ALTAI score in relation to what would be suitable to improve was also observed. This was also reported by authors in [12]. Furthermore, we received a few inquiries from the evaluators that performed the self-assessment requesting clarifications about some points of the questionnaire. This was likely due to the fact that the ALTAI tool offers guidance for each question through references to relevant parts of the Ethics Guidelines for Trustworthy AI and specific glossary. Therefore, the partners engaged in this study have defined a list of recommendations easy to use and easy to understand for other researchers, IT developers, and healthcare providers to determine the necessary level of action and urgency to tackle AI-related risks. Table 2 provides the defined list of recommendations.

## 5. Conclusions

The theoretical framework provided by this study has pointed out that AI-assisted technology in the healthcare field remains a ‘good promise,’ among others, for clinical diagnosis, improving decision making, and remote monitoring. Nevertheless, integrating AI-assisted technology in healthcare is influenced by the professionals’ and patients’ perspectives on acceptability, accountability, transparency, explainability, privacy, security, and literacy.

The ALTAI tool is a self-assessment tool for trustworthiness in AI that evaluates the rate of the AI system in seven requirements and generates a list of recommendations. This study presented the results from three SHAPES Pilots that used the ALTAI tool to self-assess the AI systems’ robustness and trustworthiness. The ALTAI category ‘transparency’ received the best results in contrast to ‘technical robustness and safety,’ which obtained the worst outcomes. This study also evaluated the functionality of this tool, which is easy to use. However, on the other side, its produced recommendations are extensive and not easy to understand and apply. Therefore, a new recommendation list was developed and validated in order to be easily considered and comprehended by the stakeholders.

## Figures and Tables

**Figure 1 healthcare-11-01454-f001:**
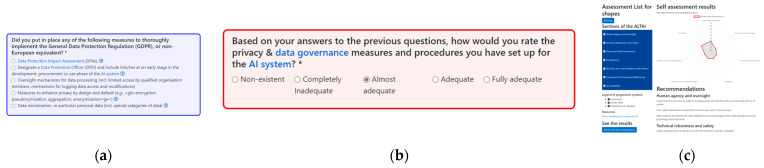
ALTAI color code: (**a**) Blue answers will contribute to recommendations (the asterisk symbol denotes that it is obligatory to answer).; (**b**) Red answers will contribute to self-assessing compliance (the asterisk symbol denotes that it is obligatory to answer); (**c**) Results and Recommendations presentation.

**Figure 2 healthcare-11-01454-f002:**
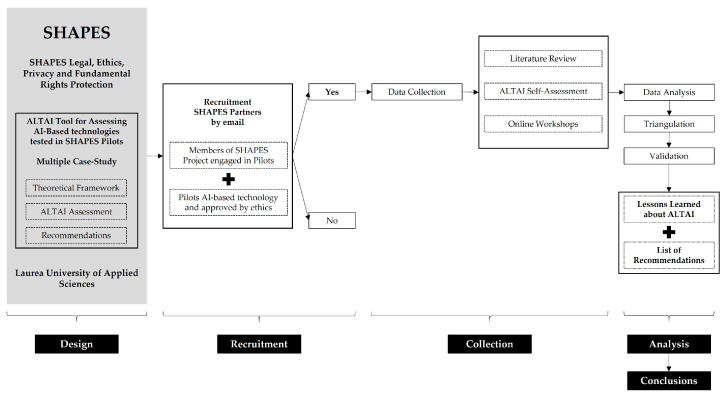
Study design.

**Figure 3 healthcare-11-01454-f003:**
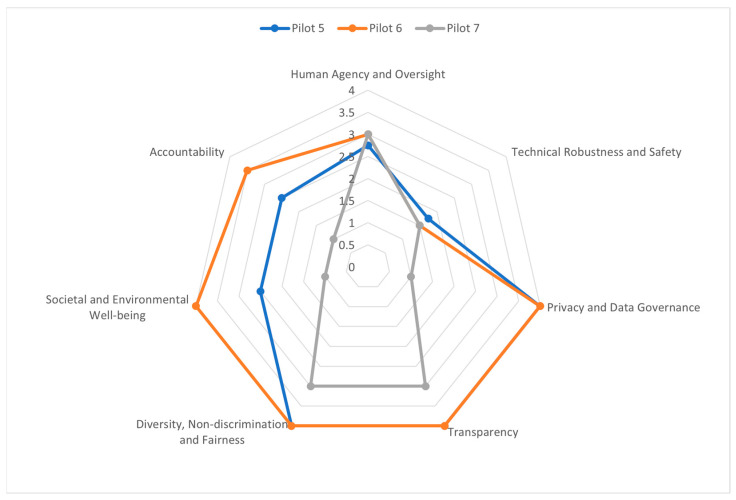
ALTAI results per SHAPES partner and cross-analysis.

**Table 1 healthcare-11-01454-t001:** SHAPES Pilots description.

	PT5 Caring for Older Individuals with Neurodegenerative Diseases	PT6 Physical Rehabilitation at Home	PT7 Cross-border Health Data Exchange
**Persona and Use Case**	Older individuals (+60 years old) living in a home. They complain about cognitive decline (e.g., daily errors, forgetting things) and require ‘attention’ from caregivers (formal and informal). The informal caregivers (e.g., children) install a digital solution in the home to monitor behaviors and health indicators, support daily activities, and provide cognitive and physical activities in the home.	Older individuals (+60 years old) living at home who either need to recover from a health issue at home (e.g., stroke, fall, surgery, rheumatoid arthritis, osteoarthritis, orofacial disorder, etc.) or need a digital solution to improve the physical activity, preventing or improving frailty conditions.	Older individuals (+60 years old) living in a community with chronic diseases (e.g., Atrial Fibrillation, Diabetes, Chronic Obstructive Pulmonary Disease, Visual or Hearing impairment, Physical Disability) who need a digital solution to improve their mobility in holidays, tourism activities, leisure, and being connected with informal caregivers and healthcare providers.
**Digital Solution**	Smart-mirror-based platform composed of hardware and software based on a smart mirror, which is equipped with a set of digital solutions (smart band, individual fall sensor, panic button, motion home sensors, cognitive and physical training programs, video call, smart agenda, and notification system).	Physical rehabilitation tool that provides a web-based platform (Phyx.io) for users and care providers. The personalized exercises prescribed in the platform are performed in front of a totem or smart mirror solution that autonomously tracks body movement and performance while doing the exercises.	Combination of smart mobile devices (smartphone, smart-band, smartwatch, tablet) connected to a healthcare platform to remote monitoring of key health parameters (heart rate, blood pressure, SPO_2_, and ECG), but also to enable healthcare providers’ remote evaluation and consultation (telemedicine).
**AI-based**	The fall detection sensor and the physical training tool use machine learning to identify certain movement patterns based on the information extracted from an Inertial Measurement Unit (IMU) and a video camera.	The physical training tool uses machine learning to identify certain movement patterns via a video camera and a pose estimation model.	Advanced filtering techniques to show a result easier to read or recognize on the screen. Data gathering methods to provide the locations with available accessibility assets. AI algorithms to identify images and detect food dishes and calories. Sensor data analysis to identify health problems.
**SHAPES Protocol**	Pilots will be assessed three times (Baseline, Post-Intervention, Follow-up) with the same protocol: WHOQOL-Bref (10 items); EQ-5D-5L (10 items); General self-efficacy scale (10 items); OSSS-3 (10 items); Participation Questionnaire (10 items); Health Literacy Measure (10 items); SUS (10 items); and Technology Acceptance Model (10 items).		

**Table 2 healthcare-11-01454-t002:** List of recommendations.

Category	Recommendations
**Human agency** **and oversight**	To avoid end-users’ full trust in AI systems. To avoid that, the system inadvertently affects human autonomy. To provide training to exercise oversight (in-the-loop, on-the-loop, in-command). To clarify all potential negative consequences for end-users or targets (e.g., develop attachments). To provide means for end-users to have control of the interactions and preserve autonomy. To have means to reduce the risk of manipulation (clear information about ownership and aims). To establish detection and response mechanisms in case of undesirable effects for the end-users. To establish control measures that reflect the self-learning/autonomous nature of the system. To involve experts from other disciplines, such as psychology and social work.
**Technical robustness** **and safety**	To assess risks of attacks to which the AI system could be vulnerable. To assess AI system threats and their consequences (design, technical, environmental, human). To assess the risk of possible malicious use, misuse, or inappropriate use of the AI system. To assess the dependency of the critical system’s decisions on its stable and reliable behavior. To control the AI system changes and its technical robustness and safety permanently.
**Privacy and data** **governance**	To adopt mechanisms that flag privacy and data protection issues. To implement the rights to withdraw consent, object, and be forgotten in the AI systems. To protect privacy and personal data during the lifecycle of an AI system (data processing). To protect non-personal data during the lifecycle of an AI system (data processing). To align the AI system with widely accepted standards (e.g., ISO) and protocols.
**Transparency**	To continuously survey users about their decisions and understanding of AI systems. To continuously assess the quality of the input data to the AI system. To explain to the end-users the AI system decisions or suggestions (answers). To explain to the end-users that AI system is an interactive machine (that he/she communicates with).
**Diversity,** **non-discrimination,** **and fairness**	To teach/educate the AI system developers about potential system bias. To implement fair AI systems and be sensitive to the variety of preferences/abilities in society. To build accessible AI systems and interfaces for all people (Universal Design principles). To assess the AI systems’ disproportional impacts considering individuals and groups. To assess the AI systems’ bias related to algorithm design (data inputs) permanently. To build algorithms that include diversity and representativeness of individuals and groups. To assess permanently the AI systems’ bias related to discrimination (e.g., race, gender, age). To adopt mechanisms to identify subjects (in) directly affected by the AI system. To adopt mechanisms that flag diversity, non-discrimination, and fairness issues. To adopt mechanisms to continuously measure the risk of bias. To provide AI systems with widely accepted definitions, concepts, and frameworks. To involve or consult the end-users in all phases of AI system development. To provide publicly available educational materials based on research and state of the art. To assess “Conflicts of Interest” of the team/individuals involved in building the AI system.
**Societal and environmental well-being**	To adopt mechanisms to identify AI systems’ positive/negative impacts on the environment. To define measures to reduce the environmental impact of AI system’s lifecycle. To involve the AI systems to tackle societal, environmental, and well-being problems. To reduce the AI systems’ negative impacts to the work and workers. To provide people with re-skill educational tools to counteract de-skilling based on AI systems. To ensure people understand the AI systems’ positive/negative impacts very well.
**Accountability**	To ensure AI systems’ auditability, modularity, and traceability (also by third parties). To fit the best practices and industry standards available and acknowledged. To ensure that all conflicts of values or tradeoffs be well-documented and explained. To include a non-technical method to assess the trustworthiness of AI (e.g., “ethical review board”). To consistently provide multisectoral and multidisciplinary auditing or guidance. To have and update the legal framework considering a wide range of impacts. To assess vulnerabilities and risks to identify and mitigate potential pitfalls.

## Data Availability

Not applicable.

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
