# Peer review of "ALTAI Tool for Assessing AI-Based Technologies: Lessons Learned and Recommendations from SHAPES Pilots"

_healthcare, 2023, doi:10.3390/healthcare11101454_

Round 1

Reviewer 1 Report

1. The paper is in fact a package of recommendation and policy statements as conclusions very suitable for an commission but completely unsuitable for a scientific paper.

2. The paper is suitable for another type of publications, maybe o commission or committee is the best solution.

Author Response

Dear Reviewer,

We would like to thank the reviewer for the fruitful comments. We have provided a revised manuscript according to the provided recommendations.

1. The paper is in fact a package of recommendation and policy statements as conclusions very suitable for a commission but completely unsuitable for a scientific paper.

Answer: Considerable improvements have been made to the manuscript, specially regarding methodology, the presentation and the analysis of the results.

2. The paper is suitable for another type of publications, maybe o commission or committee is the best solution.

Answer: The revised version of the manuscript has been reconstructed to meet the requirements of a scientific paper.

For more details please see the revised version manuscript.

Reviewer 2 Report

This paper is in an interesting and important field of healthcare research. It covers some important questions regarding the use of AI in healthcare for the elderly now, and as it develops in the future. However, unfortunately at the moment the paper doesn't reach the required standard for publication for a number of reasons, and needs further work. It is unclear what research was actually done, how it was carried out and where the results discussed have come from. The paper needs to be substantially re-written, attending to the following questions.

Why is the literature review in the results section? Lit reviews set the scene for the study that is being discussed, and the arguments that are developed there form the reasoning for why the study was done, how it was done  and what the results mean in relation to what is currently known.

What is meant by self-assessment of the pilots in the context of this paper?

What were the pilots?

Who took part?

Who did the self-assessments?

What methodology was employed for the self-assessments?

What was actually being self-assessed?

How did the cohorts that took part in the pilots differ? How were they the same?

What was actually being trialled in each of the 3 pilots?

Why is it appropriate to compare and concatenate the self-assessments from 3 different pilots?

Where do the numbers come from that are shown in the web diagram at Figure 3? It is impossible to see how this diagram matches up with the recommendations in the previous tables.

Overall, the paper needs re-working so that it is more clearly explained. I would encourage the authors to do this as it is an important area of work.

Author Response

Dear Reviewer,

We would like to thank the reviewer for the fruitful comments. We have provided a revised manuscript according to the provided recommendations.

1. This paper is in an interesting and important field of healthcare research. It covers some important questions regarding the use of AI in healthcare for the elderly now, and as it develops in the future. However, unfortunately at the moment the paper doesn't reach the required standard for publication for a number of reasons, and needs further work. It is unclear what research was actually done, how it was carried out and where the results discussed have come from. The paper needs to be substantially re-written, attending to the following questions.

Answer: Thank you for the comment. Considerable improvements have been made to reconstruct the paper according to the suggestions.

2. Why is the literature review in the results section? Lit reviews set the scene for the study that is being discussed, and the arguments that are developed there form the reasoning for why the study was done, how it was done and what the results mean in relation to what is currently known.

Answer: The literature review is, now, part of the introduction as theoretical framework. Please, see section 1 (Introduction).

3. What is meant by self-assessment of the pilots in the context of this paper?

Answer: The ALTAI tool was made to be a self-assessment tool. The tool is, now, better explain in section 2, and in the subsection related with methods.

4. What were the pilots?

Answer: More details about pilots have been included in section 3.1 of the new version of the manuscript.

5. Who took part?

Answer: More details about the pilot participants have been included in section 3.1 of the new version, especially in Table 1.

6. Who did the self-assessments?

Answer: "The leaders (coordinator) of the SHAPES Pilots selected have completed the online version ALTAI tool" (see lines 157-159, section 2.2)

7. What methodology was employed for the self-assessments?

Answer: The methodology is now described and analyzed in Section 2.2 of the revised manuscript, and resumed in Figure 2.

8. What was actually being self-assessed?

Answer: We assessed the Pilots in terms of AI trustworthy using the ALTAI tool. This is described in Section 3.1 of the new version of the manuscript and especially Table 1 provides more details about how ALTAI self-assessment was performed.

9. How did the cohorts that took part in the pilots differ? How were they the same?

Answer: In the study reported in this article, the pilots were represented just by their leaders and coordinators (one person). Nevertheless, this person represents all stakeholders engaged on the pilots’ development in SHAPES project. Authors have addressed this limitation in Section 2.3.

10. What was actually being trialed in each of the 3 pilots?

Answer: Table 1 presents now the SHAPES protocol used to trial the Pilots for their digital solution used (see section 3.1).

11. Why is it appropriate to compare and concatenate the self-assessments from 3 different pilots?

Answer: 

Thank you for the question. New section 4 and especially subsection 4.2 provides more details about the way we performed the comparison. It should be mentioned that these pilots have something in common i.e., the incorporation of AI based digital solutions to support people over 60 years old, their caregivers and their GPs. Therefore,  a triangulation analysis was developed to understand similarities and differences between the SHAPES Pilots, to validate the lessons learned and produce a comprehensive list of final ALTAI recommendations that can be used by the stakeholders.

12. Where do the numbers come from that are shown in the web diagram at Figure 3? It is impossible to see how this diagram matches up with the recommendations in the previous tables.

Answer: The ALTAI tool is an automatic tool. The spider diagram is automatically generated by the ALTAI tool see subsection 2.2. The final scores in the produced spider diagram are associated with the number of recommendations. Specifically, high score number corresponds to high number of recommendations, which are generated by the ALTAI tool. It should be mentioned that the developers of the ALTAI method have not published the way that each answer to the ALTAI questions contributes to the final scores shown in the spider diagram (subsection 4.1).

13. Overall, the paper needs re-working so that it is more clearly explained. I would encourage the authors to do this as it is an important area of work.

Answer: Thank you for your comment. The whole manuscript has been revised and restructured accordingly.

For more details please see the revised version manuscript.

Reviewer 3 Report

The authors present a framework to assess the trustworthiness of AI for technologies aimed at supporting aging in place for older adults. The paper is interesting and touches upon an important topic. I have following suggestions for improving the paper.

- Authors should clarify the origin of each ALTAI evaluation category. What were these based on or where have they been derived from? Specific references are appreciated.

- It is not clear how self-assessment was conducted. Who was involved? What kind of data was collected? How long did the self-assessment took? etc.? What kind of data did the evaluators have access to?

- The self-assessment scores are not clear. e.g., What does a 5 versus a 1 represent?

- In the discussion section, authors should clarify how researchers can use the tool in their own AI products. What kinds of limitations and advantages do they need to be aware of? What is the broader impact of this research?

Author Response

Dear Reviewer,

We would like to thank the reviewer for the fruitful comments and the guidelines on how to improve the manuscript. We have provided a revised manuscript according to the provided recommendations.

1. Authors should clarify the origin of each ALTAI evaluation category. What were these based on or where have they been derived from? Specific references are appreciated.

Answer: More details about the origin of each ALTAI evaluation category are given in subsection 2.2 of the new version of the manuscript.

2. It is not clear how self-assessment was conducted. Who was involved? What kind of data was collected? How long did the self-assessment took? etc.? What kind of data did the evaluators have access to?

Answer: Section 2 and Section 3 have been revised to include information about how self-assessment was conducted, the participants (i.e., the SHAPES pilots), their inclusion criteria and the SHAPES protocol.

3. The self-assessment scores are not clear. e.g., What does a 5 versus a 1 represent?

Answer: ALTAI tool is an automatic tool. The spider diagram is automatically generated by the ALTAI tool see subsection 2.2. The final scores in the produced spider diagram are associated with the number of recommendations. Specifically, high score number corresponds to high number of recommendations, which are generated by the ALTAI tool. It should be mentioned that the developers of the ALTAI method have not published the way that each answer to the ALTAI questions contributes to the final scores shown in the spider diagram (subsection 4.1).

4. In the discussion section, authors should clarify how researchers can use the tool in their own AI products. What kinds of limitations and advantages do they need to be aware of? What is the broader impact of this research?

Answer: Thank you for the question. New subsections 4.1 and 4.2 in the discussion provide the lessons learned from the application of the ALTAI tool. Furthermore, due to the extensive list of recommendations produced by the Tool, new recommendation list (less complexity) was generated and validated in order to be easier taken into consideration and be comprehended by the stakeholders.

For more details please see the revised version manuscript.

Round 2

Reviewer 1 Report

All the requirements are fulfilled. 

Author Response

Authors would like to thank the reviewer for the comments that assisted us to considerably improve the manuscript.

Reviewer 2 Report

Thank you for attending to my review comments so thoroughly. the paper is now much clearer and has been very much improved. Whilst it is a shame that so few SHAPES pilots were eligible to take part, I feel that the descriptions and discussions about AI assistance for the ageing population make the paper valuable and appropriate for publication. There are odd missing words etc that the manuscript needs checking for, but otherwise I am happy with this version for publication.

Author Response

We would like to thank the reviewer for the provided comments and suggestions that helped us to improve the manuscript. The manuscript has been checked for missing words and misleading sentences and was revised accordingly. The changes are highlighted with green color.

Reviewer 3 Report

The authors have addressed my comments but the English language is very confusing. In many places, sentences seem to contradict each other. 

It will be useful if authors can reference other studies that have attempted to apply AI principles to their designs. What kinds of results and conclusion have they reported?

With respect to actually doing the self-assessment, what are the evaluators feedback? Was it easy to understand and apply the questions posed in ALTAI? 

My understanding is that it becomes challenging to make changes to a system after it has been already implemented. What kind of assumptions does the ALTAI tool make about the fidelity of the system in place? In general, is it a good idea to perform a self-assessment after a high fidelity implementation?

Author Response

Dear Reviewer,

Comment 1: The authors have addressed my comments but the English language is very confusing. In many places, sentences seem to contradict each other.

Answer 1:  Authors would like to thank the reviewer for the comments that assisted us to considerably improve the manuscript. We have revised the manuscript accordingly for English language issues while misleading sentences were corrected and missing words were added. The corrections are highlighted with green color.

Comment 2: It will be useful if authors can reference other studies that have attempted to apply AI principles to their designs. What kinds of results and conclusion have they reported?

Answer 2:  Two relevant studies [11] and [12] have been included in section 1.1 Artificial Intelligence in Healthcare of the manuscript. Furthermore, in section 4.1 Lessons Learned (1st paragraph) we also discuss their reported results and conclusions “… Similar analyses [11,12] have reported the importance of ALTAI tool to provide reliable recommendations towards the improvement of trustworthiness in AI solutions. Specifically, although authors in [11] have slightly altered the ALTAI tool questions in their study, their results analysis showed that an awareness of some of the broader key areas of trustworthy factors as indicated by the ALTAI tool like accuracy of the performance of the AI solution. Moreover, the study revealed that the participated researchers were unsure whether their work is covered by the various definitions and how applicable the recommendations could be to their solutions. They recommended that a repository of ALTAI tool experiences along with good practices examples would be the key to tradeoff be-tween AI complexity and the compliance with ALTAI’s trustworthy recommendations. Similarly, the study in [12] reported that industry and public bodies should both be engaged to efficiently regulate the incorporation of ALTAI tool or its future improvements in the development of AI solutions. …”

Comment 3: With respect to actually doing the self-assessment, what are the evaluators feedback? Was it easy to understand and apply the questions posed in ALTAI?

Answer 3: Thank you for the comment. Regarding the evaluators feedback, we have included the following text to section 4.1 Lessons Learned (middle of last paragraph) “… Moreover, some of the stakeholders, involved in the self-assessments, denoted that it would be useful to know the level of the compliance with a trustworthy AI and ALTAI tool of public available machine learning models and datasets before their incorporation to their AI solutions and in order to not alter the performance of their designs. …” and to section 4.2 Recommendations (first paragraph) “… The inability of the stakeholders to interpret the ALTAI score in relation to what would be suitable to improve also was also observed. This was also reported by authors in [12].  ….”   

Comment 4: My understanding is that it becomes challenging to make changes to a system after it has been already implemented. What kind of assumptions does the ALTAI tool make about the fidelity of the system in place? In general, is it a good idea to perform a self-assessment after a high fidelity implementation?

Answer 4: Thank you for the comment. Regarding the trade-off fidelity vs. trustworthy and how the design flow of AI tools could be improved, we have included the following text to section 4.1 Lessons Learned (see last paragraph). “Furthermore, it is important to involve specific co-creation processes in the iterative design methodologies employed by most of the developers of AI solutions. Within these processes, the participation of different stakeholders such as end-users or associated institutions is considered (i.e., organizations or chambers that will use the AI technology like doctors, retail branches, etc.). Therefore, AI trustworthiness checks could be part of the co-creation activities. After the completion of each one of the associated co-creation steps, the ALTAI recommendations should be taken into account without of course de-grading the fidelity of the performance of the AI technology under development. Obviously, some of the recommendations given by the ALTAI tool should be involved from the beginning of the development procedure, as a design principle. Moreover, some of the stakeholders, involved in the self-assessments, denoted that it would be useful to know the level of the compliance with a trustworthy AI and ALTAI tool of public available machine learning models and datasets before their incorporation to their AI solutions and in order to not alter the performance of their designs. Additionally, it should be mentioned that the recommendations provided by the ALTAI tool regarding the opportunities for improvement are invaluable for organizations seeking to build a road map for the maturity of their governance [12].”

For more details please see the revised version manuscript.